# Effect of altitudes on serum parameters, metabolome, and gut microbiota in yaks on the Qinghai-Tibet Plateau

Yining Xie,[1,2,3] Yangji Cidan,[1] Zhuoma Cisang,[1] Renzeng Ciwang,[1] Guifang Liu,[1] Dan Wu,[1] Duoji Cideng,[4] Jiacuo Chilie,[4] Jilam Kang,[5] Yanbin Zhu,[1] Wangdui Basang[1]

**ABSTRACT**   Yaks (*Bos grunniens*), native to the Qinghai-Tibet Plateau, have evolved extraordinary physiological resilience to chronic hypoxia, cold, and nutritional scarcity. However, the integrated metabolic and microbial mechanisms underlying these adaptations remain poorly defined. Here, a comprehensive multi-omics analysis was performed on thirty grazing heifer yaks (2.5 years old) from three altitudes—3,600 m (low altitude [LA]), 4,000 m (middle altitude [MA]), and 4,500 m (high altitude [HA]) —to investigate how altitude affects host physiology, metabolism, and gut microbial ecology. Increasing altitude significantly reduced serum total protein, globulin, blood urea nitrogen, and alkaline phosphatase, indicating suppressed anabolic metabolism and nitrogen-sparing strategies. Antioxidant capacity (total superoxide dismutase, total antioxidant capacity) and pro-inflammatory cytokines (interleukin-2 [IL-2], IL-6, tumor necrosis factor-α, interferon-γ) increased ($P < 0.05$), while glutathione peroxidase, IL-4, IL-10, growth hormone, insulin-like growth factor-1, and growth hormone-releasing hormone declined ($P < 0.05$), reflecting energy reallocation from growth toward antioxidation and immune maintenance under hypoxia. Plasma metabolomics revealed distinct altitude-dependent reprogramming, with enrichment of retinol metabolism at 4,000 m and α-linolenic acid metabolism, tricarboxylic acid (TCA) cycle, and branched-chain amino acid biosynthesis at 4,500 m. These pathways link lipid remodeling, oxidative balance, and oxygen utilization. The gut microbiota displayed altitude-specific shifts, characterized by enrichment of *Christensenellaceae_R-7_group* and *Monoglobus* and reduced *UCG-005* and *Rikenellaceae_RC9_gut_group*, accompanied by lower fecal volatile fatty acids ($P < 0.05$). Correlation analyses confirmed tight associations between fermentative taxa and volatile fatty acids production. Collectively, our results establish a serum–metabolome–microbiota axis as a central mechanism supporting yak adaptation to high altitude.

**IMPORTANCE** This study demonstrates that the gut microbiota plays a crucial role in how yaks adapt to high-altitude hypoxia. Rising altitude not only alters the composition of gut microbes but also shifts their metabolic activity toward improving fermentation efficiency and antioxidant capacity. These microbial changes are closely linked with host metabolism, forming a coordinated serum–metabolome–microbiota network that helps maintain energy balance and immune stability when oxygen is limited. The enrichment of retinol and α-linolenic acid metabolism as altitude-responsive pathways further highlights the metabolic interplay between host and microbes in supporting physiological resilience. Overall, our findings show that microbial flexibility and metabolic cooperation are key factors enabling ruminants to survive in extreme environments, providing a scientific basis for microbiome-informed strategies to enhance yak health and productivity on the Qinghai-Tibet Plateau.

**KEYWORDS**    altitude, serum parameters, plasma metabolome, fecal microbiota

Address correspondence to Yanbin Zhu, zhuyanbin126@126.com.

Yining Xie and Yangji Cidan contributed equally to this article. Author order was determined based on their relative contributions to the study.

The authors declare no conflict of interest.

The yak (*Bos grunniens*) is a representative domestic animal inhabiting the Qinghai-Tibet Plateau and the Himalayan region, capable of surviving in extreme environments characterized by hypoxia, severe cold, and intense ultraviolet radiation (1, 2). Its milk, meat, and fiber products are vital resources for local herders, supporting both subsistence and livelihoods (3). With increasing altitude, oxygen partial pressure decreases, temperature fluctuations intensify, and ultraviolet radiation strengthens—all of which can markedly affect physiological functions, including energy metabolism, immune responses, hormonal regulation, and overall health.

Serum biochemical parameters comprehensively reflect the metabolic and physiological status of animals, including energy and protein metabolism, liver and kidney function, oxidative stress, and immune balance. These indices are valuable indicators for assessing the physiological adaptability of animals to different environmental conditions. High-altitude hypoxia not only challenges energy metabolism but also imposes persistent stress on the immune system, triggering complex immune regulatory responses (4, 5). Populations and animals living long-term in high-altitude environments often exhibit stronger immune adaptability, which is considered an evolutionary strategy to cope with environmental stress and potential pathogenic threats (6, 7). Moreover, hormones related to growth and metabolism play essential roles in regulating energy allocation, metabolic balance, and tissue development, and their variations reflect the organism's adaptive adjustments to high-altitude environments (8). In addition, rising altitude profoundly influences the composition and function of the host gut microbiota. Gut microorganisms contribute to host adaptation by modulating the production of short-chain fatty acids (VFAs) and other metabolites, thereby participating in energy metabolism and immune regulation that are crucial for maintaining host health (9, 10).

Based on these considerations, we hypothesize that with increasing altitude, yaks achieve high-altitude adaptation through the coordinated regulation of physiological metabolism, immune response, hormonal balance, and gut microbial community structure. To verify this hypothesis, this study systematically investigated serum biochemical characteristics, metabolic profiles, and gut microbial composition of yaks from different altitudes on the Qinghai-Tibet Plateau, aiming to elucidate the integrated mechanisms underlying yak adaptation to high-altitude environments. The findings will provide scientific insights into the physiological mechanisms of high-altitude adaptation in yaks and offer a theoretical basis for sustainable breeding and health management of yak populations in plateau regions.

## MATERIALS AND METHODS

### Experimental design and sample collection

The experiment was carried out in September 2024. Thirty grazing heifer yaks around 2.5 years old were randomly selected from natural pastures at an altitude of 3,600 m in Shannan City, Nedong District, Suozhu Township (N28°12′, E92°21′, low altitude group, LA), 4,000 m Linzhou County, Songpan Township (N29°45′, E90°51′, middle altitude group, MA), and 4,500 m in Nyerong County, Seqing Township (32.06°N, 92.18°E, high altitude group, HA). After all the yaks had been fasted for 24 h, tail root blood sampling was performed on them using vacuum blood collection tubes with or without heparin sodium. The samples were stored away from light for half an hour and then centrifuged at 3,000 revolutions per minute for 15 min to collect plasma and serum. Fresh feces were obtained through the rectal massage method. All the samples were temporarily stored with dry ice and then transferred to a −80°C freezer for subsequent analysis.

### Determination of serum indicators

In this study, a comprehensive analysis of biochemical indicators in serum was conducted using various diagnostic kits. The specific methods employed include the Dumas method diagnostic kit for measuring serum total protein (TP); the BCG method

diagnostic kit for measuring serum albumin; the blood urea nitrogen (BUN) method diagnostic kit for measuring BUN; the CHOD-POD method diagnostic kit for measuring serum total cholesterol (TC); the GPO-PAP method diagnostic kit for measuring triglycerides (TG); the glucose oxidase method diagnostic kit for measuring glucose; the CC kinetic method diagnostic kit for measuring serum alanine aminotransferase (ALT); the IFCC kinetic method diagnostic kit for measuring aspartate aminotransferase (AST); and the dedicated ALP detection method diagnostic kit for measuring alkaline phosphatase (ALP). Additionally, specific diagnostic kits were used to measure the levels of total superoxide dismutase (T-SOD), total antioxidant capacity (T-AOC), malondialdehyde, and glutathione peroxidase (GSH-Px) in serum. In addition, to accurately determine the levels of immune and growth-related indicators in yak serum, this study utilized bovine-specific enzyme-linked immunosorbent assay kits to quantify immunoglobulins IgA, IgM, and IgG, as well as various cytokines. These cytokines include interleukin-2 (IL-2), IL-4, IL-6, IL-10, tumor necrosis factor-α (TNF-α), interferon-γ (IFN-γ), growth hormone (GH), insulin-like growth factor-1 (IGF-1), growth hormone-releasing hormone (GHRH), and growth hormone-inhibiting hormone (GHIH).

## Plasma metabolome

Extraction of metabolites involved the addition of 50 mg of a solid sample into a 2 mL centrifuge tube that contained a 6 mm grinding bead. This was followed by the introduction of 400 μL of extraction solution (composed of methanol and water in a 4:1 ratio, with a concentration of 0.02 mg/mL). The samples were ground for 6 min at −10°C and then subjected to ultrasonication at 5°C for 30 min. Afterward, the samples were maintained at −20°C for an additional 30 min before being centrifuged at 4°C and $13,000 \times g$ for 15 min. The resulting supernatant was subsequently placed into an injection vial for analysis via LC-MS/MS. A pooled quality control (QC) sample was generated by combining equal volumes from all individual samples, which was injected regularly to ensure the stability of the analytical results. The LC-MS/MS analysis utilized a Thermo UHPLC-Exploris 240 system equipped with an ACQUITY HSS T3 column. The mobile phases included 0.1% formic acid in water and acetonitrile (95:5), as well as 0.1% formic acid in a mixture of acetonitrile, isopropanol, and water (47.5:47.5:5). Gradient elution was carried out with varying profiles for both positive and negative ion modes, while the mass spectrometric conditions were fine-tuned using an ESI source, with adjustments made to temperature, gas flow rates, and voltages. Data acquisition was executed in Data Dependent Acquisition mode within a mass range of 70–1,050 m/z.

## Fecal volatile fatty acids

A stock solution A was prepared by mixing six VFAs—acetic, propionic, butyric, isobutyric, isovaleric, and valeric—in 9,840 μL of HPLC-grade butanol. Stock solution B was created by adding 10 μL of 2-ethylbutyric acid to 9,990 μL of butanol as an internal standard. These stock solutions were further diluted into seven working solutions for GC-MS analysis. For sample preparation, 25 mg of fecal samples were processed by adding 500 μL of water containing 0.5% phosphoric acid. The samples were then subjected to freeze-grinding (50 Hz, 3 min × 2), sonication (10 min), and centrifugation (4°C, $13,000 \times g$, 15 min). The supernatant (200 μL) was extracted with 200 μL of butanol containing 10 μg/mL 2-ethylbutyric acid, followed by vortexing (10 s), sonication (10 min), and another centrifugation (4°C, $13,000 \times g$, 5 min). The final supernatant was transferred to vials for GC-MS analysis. The GC-MS analysis was performed using an Agilent 8890B-5977B/7000D instrument equipped with an HP FFAP column and helium as the carrier gas. The temperature program was set to start at 80°C, increase to 120°C at a rate of 20°C/min, then to 160°C at 5°C/min, and finally hold at 220°C for 3 min. The mass spectrometry conditions included an EI ion source with temperatures set at 230°C for the ion source and 150°C for the quadrupole, and data acquisition was performed in SIM mode. To ensure system stability and repeatability, QC samples were inserted every

10 samples, with relative standard deviation (RSD) values for target compounds required to be below 15%.

## Fecal microbiota

Genomic DNA in 30 fecal samples was isolated from yak fecal using the E.Z.N.A. Soil DNA Kit (Omega Bio-tek, Norcross, GA, USA) and checked via 1% agarose gel electrophoresis (8). The V3–V4 region of the bacterial 16S rRNA gene was then amplified with primers 338F (5′-ACTCCTACGGGAGGCAGCAG-3′) and 806R (5′-GGACTACHVGGGTWTCTAAT-3′). The amplicons were sequenced on the Illumina MiSeq PE300 platform. Data analysis was performed using Uparse (v7.0) to cluster sequences into operational taxonomic units (OTUs) at a 97% similarity threshold and remove chimeric sequences. Taxonomic classification was conducted using the Ribosomal Database Project Classifier (v2.11) with the 16S rRNA database, applying a confidence threshold of 0.7.

## Data analysis

The data matrix was uploaded to the Majorbio Cloud Platform for analysis. It was preprocessed by removing variables with missing values using the 80% rule; imputing missing values with the minimum value; normalizing with total sum normalization; removing variables with RSD >30% in QC samples; and log10-transforming the data to obtain the final matrix for analysis. Principal component analysis (PCA) and orthogonal partial least squares discriminant analysis (OPLS-DA) were performed using the ropls package in R, with sevenfold cross-validation to assess model stability. To elucidate the metabolic pathways in our study, we utilized the Kyoto Encyclopedia of Genes and Genomes (KEGG) Pathway Database (11). This database offers detailed and comprehensive information on metabolic pathways and associated biological processes. The relevant data were obtained from the KEGG database (https://www.kegg.jp/) and subsequently processed as detailed in the subsequent sections. Differential metabolites were identified based on VIP >1 from the OPLS-DA model and $P < 0.05$ from Student's $t$-test. Pathway analysis was conducted using the KEGG database, and pathway enrichment was performed with scipy stats in Python, identifying relevant biological pathways through Fisher's exact test. The bioinformatics analysis was conducted using Usearch software through the Majorbio Cloud Platform, clustering OTUs at a 97% similarity level for statistical analysis. Alpha diversity (Ace, Chao1, Shannon) was calculated using Mothur, and beta diversity (PCoA) was analyzed with Qiime 2. Microbial abundances at phylum and genus levels were compared using R and Python packages (Wilcoxon rank sum test). Differential taxa were identified using linear discriminant analysis effect size (LEfSe) with a Kruskal-Wallis alpha value of 0.05 and an LDA score threshold of 3.5. A Pearson correlation analysis was performed using the Pheatmap package in R, with significance set at $P \leq 0.05$.

Before conducting statistical analysis, the normality and homogeneity of variance of the data were checked using the test procedures of JMP software (JMP, version 10; SAS Institute Inc., Cary, NC). If $P < 0.05$, it is considered significant, and if $0.05 \leq P \leq 0.10$, it may indicate a trend.

## RESULTS

### Serum biochemical indicators

Compared with the low altitude group, as the altitude increased, the levels of total TP, GLB, BUN, ALT, and ALP in the serum of yaks in the MA group and the HA group decreased significantly ($P < 0.05$, Table 1). Meanwhile, the levels of TG in the serum of yaks in the LA group and the MA group were significantly lower than those in the HA group ($P < 0.05$). Interestingly, the MA group had significantly higher TC but lower AST levels compared to the other two groups ($P < 0.05$).

## Serum antioxidant indicators

When compared with the low-altitude group, as the altitude increased, the levels of T-SOD and T-AOC in the serum of yaks in both the MA group and the HA group increased significantly ($P < 0.05$, Table 2). Conversely, the enzymatic activity of GSH-Px decreased significantly ($P < 0.05$).

## Serum immune indicators

With increasing altitude, the levels of immunoglobulins (IgA, IgM, and IgG), interleukins (IL-2, IL-6, and TNF-α), and IFN-γ in the serum of yaks in the MA and HA groups increased significantly ($P < 0.05$; Table 3). In contrast, the levels of IL-4 and IL-10 decreased significantly in the MA and HA groups ($P < 0.05$).

## Serum hormonal indicators

Compared with the LA group, the levels of GH, IGF-1, and GHRH in the serum of yaks in the MA and HA groups decreased significantly ($P < 0.05$; Table 4). Meanwhile, the concentration of GHIH increased significantly with increasing altitude ($P < 0.05$).

## Fecal volatile fatty acids

As shown in Table 5, the concentrations of acetic acid, propionic acid, isobutyric acid, butyric acid, isovaleric acid, and valeric acid in the feces of yaks in the MA and HA groups were significantly lower than those in the LA group ($P < 0.05$).

## Plasma metabolome

PCA revealed clear separations in the plasma metabolic profiles of yaks from different altitudes (R = 0.92, $P < 0.01$), demonstrating the scientific validity and reliability of the subsequent analyses (Fig. 1A). Venn diagram analysis showed that 1,804, 1,852, and 1,845 metabolites were detected in the plasma of yaks from the LA group, MA group, and HA group, respectively, with 1,686 metabolites shared across all three groups (Fig. 1B). Compared with the LA group, 414 metabolites were upregulated and 214 were downregulated in the MA group, whereas 455 metabolites were upregulated and 191 were downregulated in the HA group (Fig. 1C). Pathway enrichment analysis further identified the top five significantly enriched pathways in the MA group, including *retinol metabolism*, *tryptophan metabolism*, *nucleotide metabolism*, *glycerophospholipid metabolism*, and *purine metabolism* (Fig. 1D). In contrast, the top five enriched pathways in the HA group were *α-linolenic acid metabolism*, *citrate cycle (TCA cycle)*, *valine, leucine, and isoleucine biosynthesis*, *phenylalanine, tyrosine, and tryptophan biosynthesis*, and *tyrosine metabolism* (Fig. 1E).

**TABLE 1** Effect of altitude on biochemical indicators of yak serum[b]

| Items | Treatment[a] | | | SEM | P value |
|---|---|---|---|---|---|
| | LA | MA | HA | | |
| Total protein, g/L | 74.66[a] | 70.96[a] | 63.39[b] | 2.42 | 0.011 |
| Albumin, g/L | 43.35 | 44.65 | 40.42 | 1.41 | 0.1171 |
| Globulin, g/L | 30.80[a] | 26.68[b] | 24.15[b] | 1.27 | 0.0047 |
| Urea nitrogen, mmol/L | 6.64[a] | 4.76[b] | 4.15[b] | 0.29 | <0.0001 |
| Total cholesterol, mmol/L | 2.86[b] | 4.66[a] | 2.58[b] | 0.33 | 0.0004 |
| Triglycerides (TG), mmol/L | 1.36[b] | 1.30[b] | 1.49[a] | 0.03 | 0.0004 |
| Glucose, mmol/L | 8.13 | 4.79 | 4.14 | 0.32 | 0.1065 |
| Alanine aminotransferase, U/L | 108.33[a] | 93.38[b] | 79.88[c] | 2.84 | <0.0001 |
| Aspartate aminotransferase, U/L | 41.43[a] | 26.68[b] | 39.63[a] | 3.01 | 0.0042 |
| Alkaline phosphatase, U/L | 166.88[a] | 155.25[a] | 117.88[b] | 11.72 | 0.0196 |

[a]LA, low altitude yak group; MA, middle altitude yak group; HA, high altitude yak group, $n = 10$.
[b]a–c: values within a row that do not share a common superscript differ significantly ($P < 0.05$).

**TABLE 2** Effect of altitude on antioxidant indicators of yak serum[b]

| Items | Treatment[a] | | | SEM | P value |
|---|---|---|---|---|---|
| | LA | MA | HA | | |
| Total superoxide dismutase, U/mL | 74.66[a] | 70.96[a] | 63.39[b] | 2.42 | 0.011 |
| Total antioxidant capacity, U/mL | 108.33[a] | 93.38[b] | 79.88[c] | 2.84 | <0.0001 |
| Malondialdehyde, nmol/mL | 41.43[a] | 26.68[b] | 39.63[a] | 3.01 | 0.0042 |
| Glutathione peroxidase, U/mL | 166.88[a] | 155.25[a] | 117.88[b] | 11.72 | 0.0196 |

[a]LA, low altitude yak group; MA, middle altitude yak group; HA, high altitude yak group, $n = 10$.
[b]a–c: values within a row that do not share a common superscript differ significantly ($P < 0.05$).

## Fecal microbiota

A total of 30 fecal samples from yaks across three altitude groups were analyzed to assess microbial diversity. Altitude significantly affected the α-diversity of the fecal microbiota (Fig. 2A through C), as evidenced by a reduced Shannon index in the MA group and lower Chao1 and Shannon indices in the HA group. In addition, altitude markedly influenced the β-diversity of the fecal microbiota, as indicated by distinct PCoA clustering among groups (R = 0.56, $P$ = 0.001; Fig. 2D). The microbial dysbiosis index (MDI) also showed a significant elevation in the MA and HA groups compared with the LA group (Fig. 2E), suggesting increased microbial imbalance with rising altitude. At the phylum level, altitude significantly decreased the relative abundance of Verrucomicrobiota (Fig. 2F), as well as Actinobacteriota in the HA group and Actinobacteriota in both the MA and HA groups (Fig. S1A and B). At the genus level, among the top 20 taxa (Fig. 2G), 13 genera exhibited significant differences across altitudes (Fig. S1C through O). Compared with the LA group, the MA group showed a higher relative abundance of *UCG-005*, while *norank_f__Eubacterium_coprostanoligenes_group*, *Christensenellaceae_R-7_group*, *Monoglobus*, *norank_f__UCG-010*, *Prevotellaceae_UCG-004*, and *NK4A214_group* were significantly reduced. In the HA group, the relative abundances of *UCG-005*, *Rikenellaceae_RC9_gut_group*, *Alistipes*, and *norank_f__Muribaculaceae* were significantly increased, whereas *norank_f__Eubacterium_coprostanoligenes_group*, *Christensenellaceae_R-7_group*, *Monoglobus*, *Akkermansia*, *Prevotellaceae_UCG-004*, *norank_f__Ruminococcaceae*, *Romboutsia*, and *NK4A214_group* were significantly decreased. LEfSe analysis (LDA score > 3) identified 34 differential taxa ranging from phylum to genus level among the three altitude groups (Fig. 3). Functional prediction based on PICRUSt2 indicated that the altered fecal microbiota were mainly enriched in metabolic pathways, biosynthesis of secondary metabolites, biosynthesis of amino acids, microbial metabolism in diverse environments, and carbon metabolism (Fig. 4A). Moreover, BugBase phenotypic prediction revealed that, relative to the LA group, the HA group showed significantly lower proportions of microorganisms classified as Contains Mobile Elements

**TABLE 3** Effect of altitude on immune indicators of yak serum[b]

| Items | Treatment[a] | | | SEM | P value |
|---|---|---|---|---|---|
| | LA | MA | HA | | |
| Immunoglobulin A, g/L | 1.07[c] | 1.19[b] | 1.28[a] | 0.01 | <0.0001 |
| Immunoglobulin M, g/L | 0.90[c] | 1.00[b] | 1.08[a] | 0.01 | <0.0001 |
| Immunoglobulin G, g/L | 4.74[c] | 5.41[b] | 5.82[a] | 0.05 | <0.0001 |
| Interleukin-2, pg/mL | 183.70[c] | 233.96[b] | 286.78[a] | 3.60 | <0.0001 |
| Interleukin-4, pg/mL | 74.59[a] | 65.23[b] | 55.24[c] | 0.67 | <0.0001 |
| Interleukin-6, pg/mL | 99.11[c] | 138.20[b] | 173.65[a] | 3.38 | <0.0001 |
| Interleukin-10, pg/mL | 257.27[a] | 214.14[b] | 177.75[c] | 2.59 | <0.0001 |
| Tumor necrosis factor α, pg/mL | 109.50[c] | 139.45[b] | 171.39[a] | 2.65 | <0.0001 |
| Interferon-γ, pg/mL | 870.38[c] | 1,134.49[b] | 1,437.88[a] | 22.18 | <0.0001 |

[a]LA, low altitude yak group; MA, middle altitude yak group; HA, high altitude yak group, $n = 10$.
[b]a–c: values within a row that do not share a common superscript differ significantly ($P < 0.05$).

**TABLE 4** Effect of altitude on growth hormone axis indicators of yak serum[b]

| Items | Treatment[a] | | | SEM | P value |
|---|---|---|---|---|---|
| | LA | MA | HA | | |
| Growth hormone, ng/mL | 9.77[a] | 7.68[b] | 6.10[c] | 0.23 | <0.0001 |
| Insulin-like growth factor-1, ng/mL | 231.55[a] | 185.94[b] | 147.12[c] | 5.60 | <0.0001 |
| Growth hormone-releasing hormone, pg/mL | 24.51[a] | 20.88[b] | 16.86[c] | 0.42 | <0.0001 |
| Growth hormone-inhibiting hormone, pg/mL | 122.38[c] | 153.63[b] | 193.68[a] | 4.30 | <0.0001 |

[a]LA, low altitude yak group; MA, middle altitude yak group; HA, high altitude yak group, $n = 10$.
[b]a–c: values within a row that do not share a common superscript differ significantly ($P < 0.05$).

and Aerobic, but significantly higher proportions of Anaerobic and Potentially Pathogenic phenotypes (Fig. 4B through F).

## Correlation analysis

Spearman's correlation analysis was conducted between six VFAs and the top 20 microbial genera(Fig. 5). The results showed that acetic acid was positively correlated with *NK4A214_group* (R = 0.37, *P* = 0.05) and *Christensenellaceae_R-7_group* (R = 0.39, *P* = 0.03). Propanoic acid exhibited positive correlations with *NK4A214_group* (R = 0.40, *P* = 0.03), *Monoglobus* (R = 0.37, *P* = 0.04), and *norank_f__Eubacterium_coprostanoligenes_group* (R = 0.38, *P* = 0.04). Isobutyric acid showed significant positive correlations with *NK4A214_group* (R = 0.63, *P* < 0.01), *norank_f__Ruminococcaceae* (R = 0.42, *P* = 0.02), *Akkermansia* (R = 0.49, *P* = 0.01), *norank_o__Clostridia_UCG-014* (R = 0.48, *P* = 0.01), and *Monoglobus* (R = 0.56, *P* < 0.01), while it was negatively correlated with *norank_f__Muribaculaceae* (R = −0.51, *P* < 0.01), *unclassified_f__Lachnospiraceae* (R = −0.48, *P* = 0.01), and *Rikenellaceae_RC9_gut_group* (R = −0.38, *P* = 0.04). Butanoic acid was positively correlated with *NK4A214_group* (R = 0.47, *P* = 0.01), *Monoglobus* (R = 0.38, *P* = 0.04), and *Christensenellaceae_R-7_group* (R = 0.45, *P* = 0.01). Similarly, isovaleric acid exhibited positive correlations with *NK4A214_group* (R = 0.58, *P* < 0.01), *norank_f__Ruminococcaceae* (R = 0.38, *P* = 0.04), *Akkermansia* (R = 0.40, *P* = 0.03), *norank_o__Clostridia_UCG-014* (R = 0.39, *P* = 0.03), and *Monoglobus* (R = 0.50, *P* = 0.01), but negative correlations with *norank_f__Muribaculaceae* (R = −0.49, *P* = 0.01), *Alistipes* (R = −0.39, *P* = 0.03), and *unclassified_f__Lachnospiraceae* (R = −0.38, *P* = 0.04). In addition, valeric acid was positively correlated with *NK4A214_group* (R = 0.52, *P* < 0.01), *Monoglobus* (R = 0.49, *P* = 0.01), and *norank_f__Eubacterium_coprostanoligenes_group* (R = 0.45, *P* = 0.01).

## DISCUSSION

This study provides a comprehensive multi-omics framework to elucidate how yaks orchestrate physiological, metabolic, and microbial adaptations to cope with the hypoxic and nutritional challenges of high-altitude environments. By integrating serum biochemistry, antioxidant and immune parameters, endocrine factors, plasma metabolomics, and gut microbiota profiles across the LA, MA, and HA groups, we demonstrate that altitude triggers a coordinated reprogramming of host metabolism and microbial

**TABLE 5** Effect of altitude on fecal volatile fatty acids of yak[b]

| Items | Treatment[a] | | | SEM | P value |
|---|---|---|---|---|---|
| | LA | MA | HA | | |
| Acetic acid, ug/mg | 2,983.43[a] | 1,241.95[b] | 1,380.66[b] | 198.21 | <0.0001 |
| Propanoic acid, ug/mg | 687.00[a] | 242.34[b] | 268.23[b] | 47.86 | <0.0001 |
| Isobutyric acid, ug/mg | 103.27[a] | 66.84[b] | 51.66[b] | 8.99 | 0.0012 |
| Butanoic acid, ug/mg | 472.32[a] | 179.15[b] | 171.38[b] | 39.83 | <0.0001 |
| Isovaleric acid, ug/mg | 43.65[a] | 25.36[b] | 20.21[b] | 4.72 | 0.0041 |
| Valeric acid, ug/mg | 79.35[a] | 30.61[b] | 29.60[b] | 6.60 | <0.0001 |

[a]LA, low altitude yak group; MA, middle altitude yak group; HA, high altitude yak group, $n = 10$.
[b]a–c: values within a row that do not share a common superscript differ significantly ($P < 0.05$).

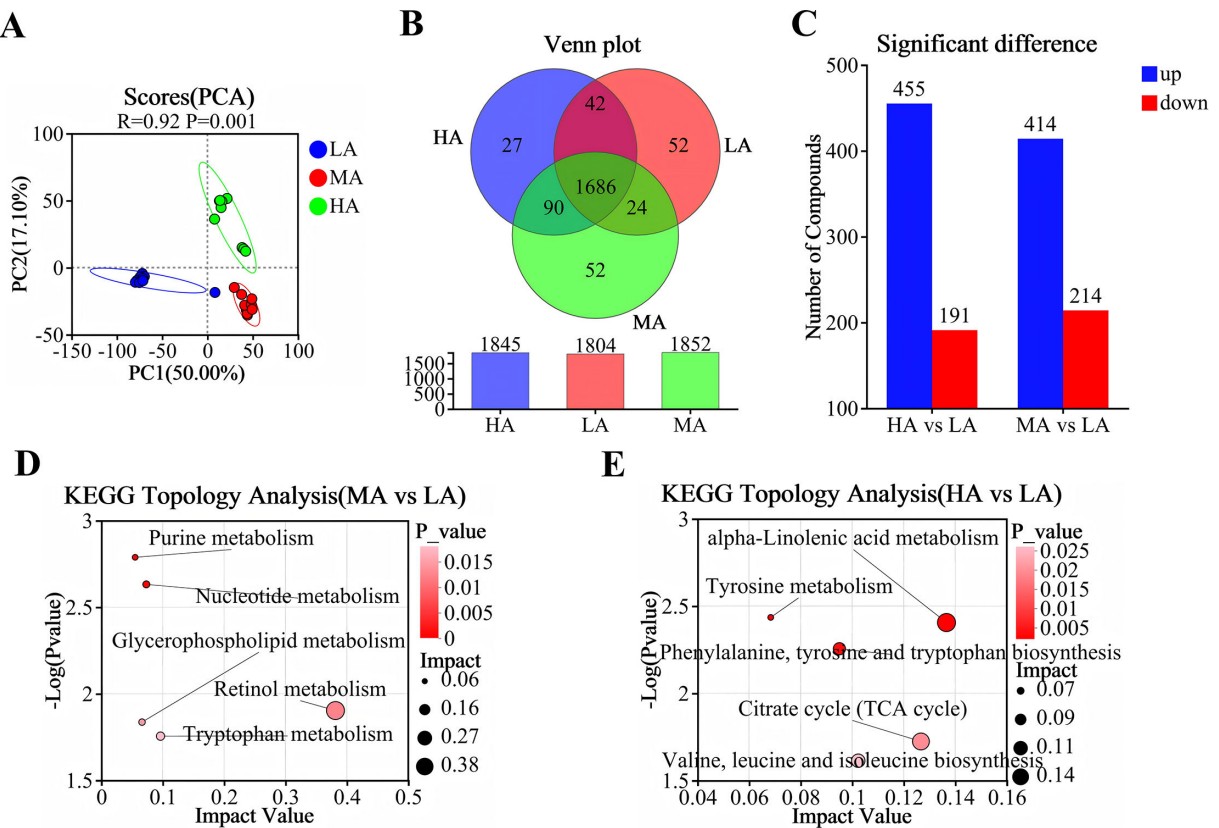

FIG 1 Plasma metabolome analysis diagrams. (A) PCA score plot; (B) Venn plot; (C) volcano plot of differential metabolites (MA vs LA); (D) KEGG topological analysis diagram (MA vs LA); (E) volcano plot of differential metabolites (HA vs LA); KEGG topological analysis diagram (HA vs LA). LA, low altitude yak group; MA, middle altitude yak group; HA, high altitude yak group, *n* = 10.

ecology. The results support an adaptive model of energy reallocation under hypoxic stress, whereby yaks shift resources from growth and protein anabolism toward maintenance, antioxidation, and immune defense to sustain homeostasis in extreme environments.

Serum biochemical indicators provide a sensitive reflection of the metabolic and nutritional status of yaks living under environmental stress (12). In our study, yaks at HA exhibited significantly lower serum total protein, ALP, and BUN, consistent with attenuated anabolic metabolism and limited nutrient intake at high altitude (13, 14). The decline in BUN—a byproduct of amino acid catabolism—indicates a nitrogen-sparing strategy aimed at reducing protein degradation and conserving nitrogen under dietary restriction (15). Similarly, the decrease in ALT activity across MA and HA suggests impaired hepatic amino acid metabolism and reduced protein synthesis efficiency (16). Collectively, these alterations indicate that yaks at higher altitudes reduce anabolic metabolism to prioritize essential maintenance functions over growth. The endocrine data reinforce this interpretation. Levels of GH, IGF-1, and GHRH decreased progressively with altitude, while GHIH increased, reflecting suppression of the GH/IGF-1 axis under hypoxic conditions. This endocrine suppression is consistent with the energy reallocation hypothesis, in which animals redirect metabolic energy from growth to survival-related processes, such as oxygen transport and stress resistance (17). GH and IGF-1 are pivotal regulators of growth, nutrient metabolism, and oxidative balance (18, 19); their downregulation, coupled with enhanced inhibitory signaling, represents a strategic adaptation to limited energy availability in the HA environment. Similar patterns have been reported in high-altitude mammals, where hypoxia induces cortisol secretion, gluconeogenesis, and inhibition of growth-related pathways (17). Thus, the endocrine

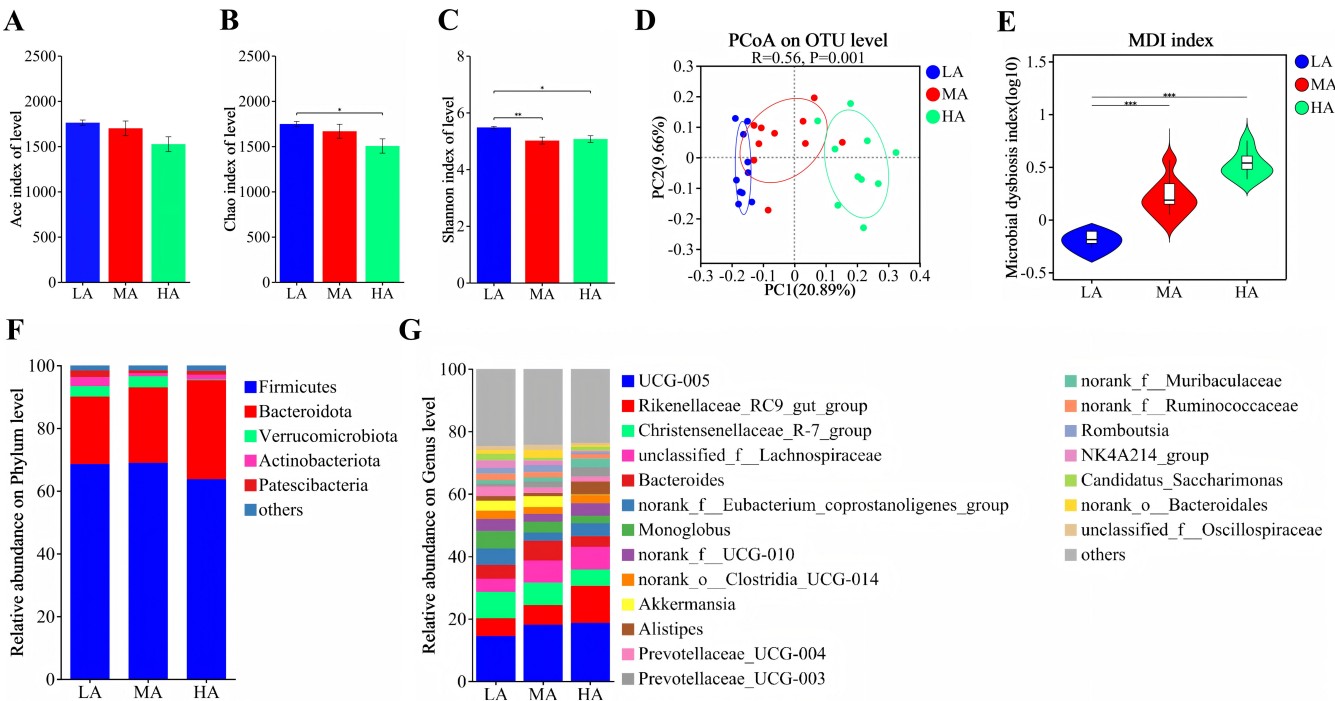

**FIG 2** The diversity of fecal microbiota. (A) Ace index; (B) *Chao1* index; (C) Shannon index; (D) PCoA score plot; (E) MDI; (F) microbial compositions at the phylum level; (G) microbial compositions at the genus level. *, ** and *** indicate $P < 0.05$, $P < 0.01$, and $P < 0.001$, respectively.

remodeling observed here exemplifies a trade-off between growth and homeostasis, a hallmark of chronic hypoxic adaptation.

High-altitude hypoxia increases mitochondrial ROS production during intermittent reoxygenation, imposing oxidative stress on tissues (20). Our data revealed altitude-dependent modulation of the antioxidant defense system. In the MA and HA groups, T-SOD and T-AOC were significantly elevated, whereas GSH-Px activity declined. This pattern suggests that oxidative stress intensifies with altitude, prompting upregulation of primary antioxidant defenses (SOD, T-AOC) while secondary defenses (GSH-Px) may be constrained by selenium deficiency or excessive utilization (21, 22). Enhanced T-SOD and T-AOC activity help neutralize superoxide radicals, maintaining cellular redox balance and protecting lipids and proteins from peroxidation damage (20). Such enzyme adjustments have been widely observed in plateau animals as adaptive strategies to mitigate hypoxia- and UV-induced oxidative injury (4). Increased antioxidant activity at high altitude indicates that yaks possess an intrinsic redox flexibility, allowing them to fine-tune enzyme systems to maintain redox homeostasis despite fluctuating oxygen availability. The decrease in GSH-Px may represent a compensatory redistribution of enzymatic roles, where SOD assumes greater importance in detoxifying superoxide radicals. Together, these findings suggest that antioxidant enhancement is an essential component of yak adaptation, enabling long-term survival under oxidative pressure in the Qinghai-Tibet Plateau.

The immune system is another key target of altitude adaptation, balancing pro-inflammatory activation for defense with anti-inflammatory regulation to prevent tissue damage. In the MA and HA groups, serum levels of immunoglobulins (IgA, IgM, IgG) and pro-inflammatory cytokines (IL-2, IL-6, TNF-α, IFN-γ) were significantly elevated, while anti-inflammatory cytokines (IL-4, IL-10) declined. Elevated immunoglobulins reflect enhanced humoral immune responses, possibly as protection against cold stress and hypoxia-related infections (4, 6). The simultaneous rise in IL-2, IL-6, TNF-α, and IFN-γ indicates activation of Th1-type immune pathways and macrophage function (23, 24), which strengthen host defense and tissue repair. Conversely, the suppression of IL-4 and

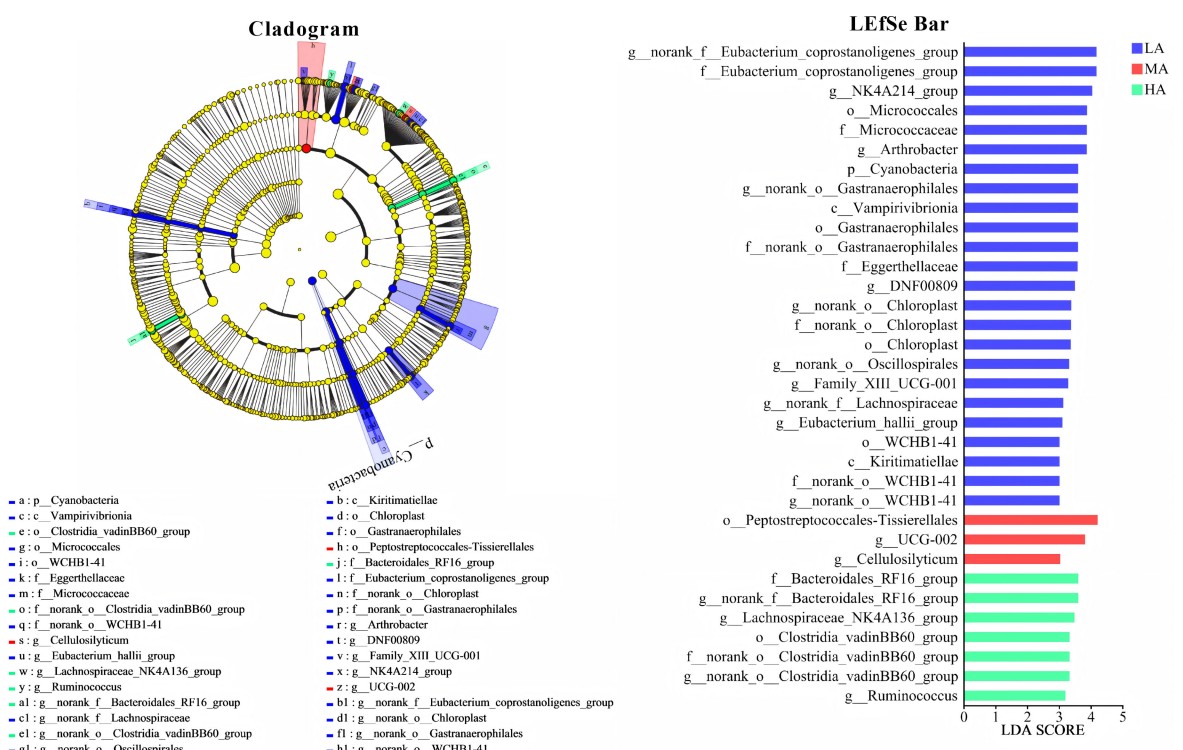

**FIG 3** LEfSe hierarchical taxonomic tree at multiple levels.

IL-10 suggests reduced anti-inflammatory control. Under severe environmental stress, the body may transiently favor pro-inflammatory responses to eliminate pathogens and repair hypoxia-induced tissue injury (25, 26). IL-6, in particular, serves dual roles in inflammation and erythropoiesis by activating HIF-1/EPO signaling (27, 28), contributing to improved oxygen transport. Therefore, this cytokine profile represents a regulated pro-inflammatory state characteristic of highland adaptation. Although prolonged inflammation could be harmful, long-term high-altitude residents, such as yaks, likely achieve equilibrium between immune activation and tolerance. This immune rebalancing ensures effective immune surveillance without triggering chronic inflammation.

Untargeted plasma metabolomics revealed significant altitude-driven enrichment of multiple metabolic pathways. At MA, retinol metabolism was the most prominent pathway, suggesting its involvement in redox regulation and immune modulation under moderate hypoxia. Retinol and retinoic acid maintain epithelial integrity, regulate antioxidant gene expression, and modulate HIF signaling (29, 30). Enhanced retinol metabolism may therefore reinforce antioxidant defense and mucosal protection, helping yaks sustain tissue function in oxygen-poor environments (31). At HA, α-linolenic acid (ALA) metabolism and the TCA cycle were notably enriched. ALA, an essential omega-3 fatty acid, contributes to membrane fluidity, anti-inflammatory responses, and efficient energy production (9, 32). Its increased metabolism suggests adaptive lipid remodeling to sustain mitochondrial performance under hypoxia. Meanwhile, enrichment of the TCA cycle indicates intensified flux through core energy pathways, potentially reflecting accumulation of intermediates, such as succinate, which stabilizes HIF-1α and activates hypoxia-response genes (32, 33). Enhanced TCA and branched-chain amino acid (valine, leucine, isoleucine) metabolism may improve ATP generation efficiency and support nitrogen balance under oxygen limitation (9, 34). These findings demonstrate a metabolic flexibility that enables yaks to switch between lipid oxidation, amino acid turnover, and glycolysis to maintain energetic homeostasis across altitudes. Notably, alterations in glycosaminoglycan biosynthesis (heparan sulfate/heparin) were detected, indicating adjustments in cell signaling, tissue repair, and extracellular matrix

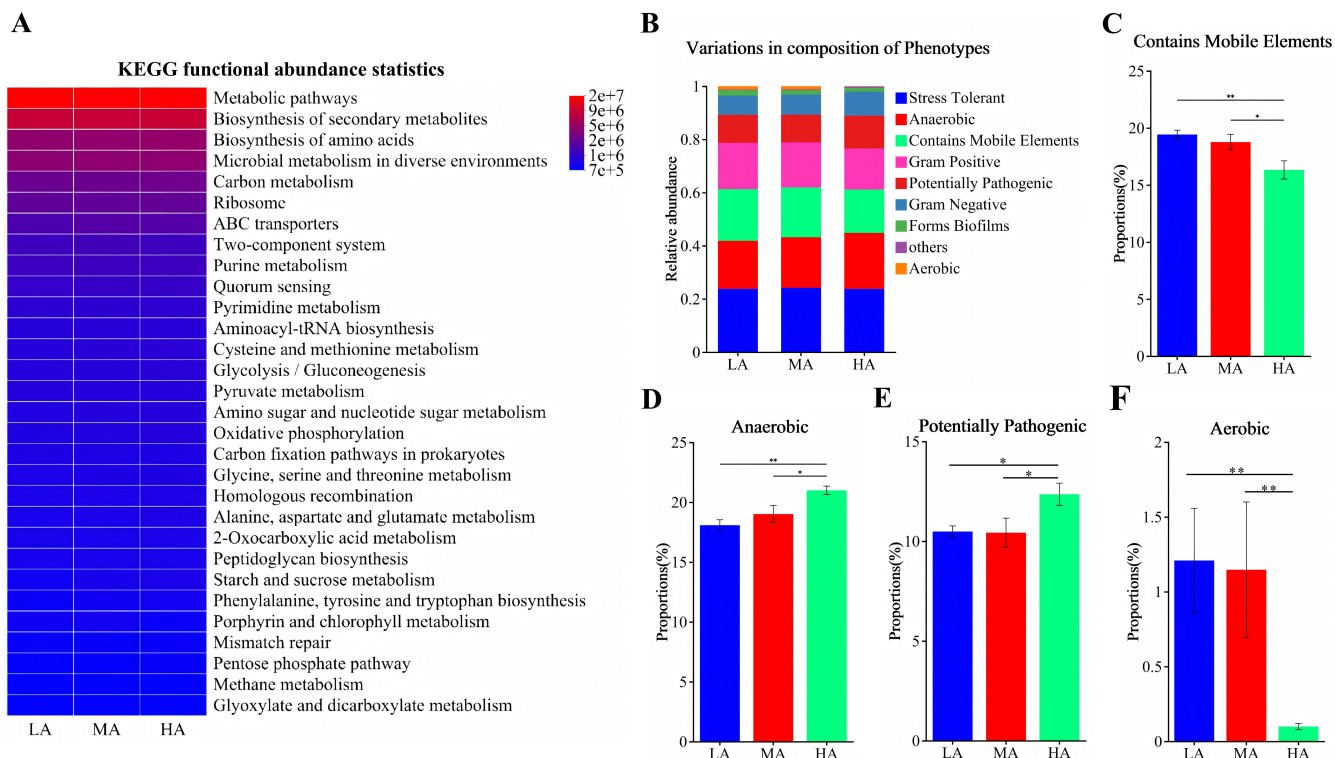

**FIG 4** Functional prediction of microbial communities. (A) Functional prediction based on PICRUSt2; (B) phenotypic prediction using BugBase; (C) relative abundance of *Contains Mobile Elements* phenotype; (D) relative abundance of *Anaerobic* phenotype; (E) relative abundance of *Potentially Pathogenic* phenotype; and (F) relative abundance of *Aerobic* phenotype. * and ** indicate $P < 0.05$ and $P < 0.01$, respectively.

remodeling (35, 36). These modifications may enhance vascular stability and protect against hypoxia-induced tissue injury, further supporting physiological adaptation. Overall, the metabolomic reprogramming of yaks reveals an integrated network linking antioxidant defense, energy metabolism, and oxygen signaling, enabling resilience to extremely high-altitude conditions.

The gut microbiota plays a crucial role in nutrient acquisition and immune regulation, and its composition was profoundly shaped by altitude. 16S rRNA analysis revealed that microbial α-diversity (*Chao1*, Shannon) and fermentative capacity (VFA concentrations) declined with altitude, indicating a less diverse and functionally constrained community. Such reductions reflect environmental filtering by hypoxia, cold, and limited forage diversity (9, 37). The MDI increased in the MA and HA groups, suggesting decreased microbial stability and a shift toward stress-tolerant taxa. Taxonomic analysis revealed altitude-dependent shifts in key microbial groups. *UCG-005* and *Rikenella-ceae_RC9_gut_group*, belonging to Bacteroidetes, declined with altitude, potentially due to reduced availability of plant polysaccharides (37, 38). In contrast, *Christensenel-laceae_R-7_group* and *Monoglobus* increased, indicating selective enrichment of taxa associated with host metabolic efficiency and fiber degradation (17, 39). *Christensenella-ceae* has been linked to host leanness, metabolic health, and heritable microbiome traits, while *Monoglobus* specializes in pectin utilization, supporting carbohydrate fermentation (39). The rise of *norank_f__Ruminococcaceae* in HA yaks likely enhances VFA synthe-sis, compensating partially for reduced fermentative output. Consistent with these compositional changes, fecal VFAs (acetate, propionate, butyrate) declined significantly with altitude, indicating diminished fiber fermentation and energy harvest (40). VFAs are critical energy substrates for ruminants and also contribute to intestinal barrier function and immune regulation (41). The observed decline may thus contribute to lower nutrient absorption and altered immune homeostasis at high altitude. Moreover,

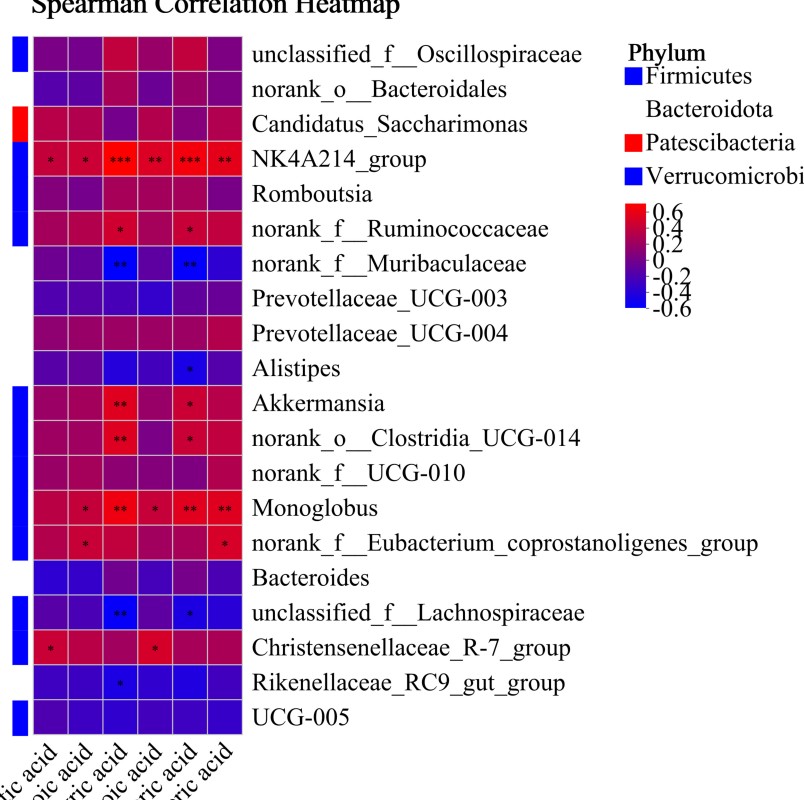

## Spearman Correlation Heatmap

**FIG 5** Spearman's correlation analysis between VAFs and the top 20 microbial genera. *, **, and *** indicate $P < 0.05$, $P < 0.01$, and $P < 0.001$, respectively.

BugBase functional prediction revealed a decrease in anaerobic taxa and an increase in mobile genetic elements, suggesting microbial genomic adaptation to environmental stress.

Spearman correlation analysis further revealed tight associations among plasma metabolites, fecal VFAs, and dominant microbial genera. In particular, acetic and propionic acids were positively correlated with *Christensenellaceae_R-7_group* and *Monoglobus*, whereas *Rikenellaceae_RC9_gut_group* showed a negative association. These relationships suggest that beneficial fermenters contribute to efficient fiber degradation and energy extraction, while altitude-enriched taxa may reflect metabolic stress or alternative fermentation routes. Similar metabolite–microbiota correlations have been reported in other highland ruminants, where specific taxa drive short-chain fatty acid production and energy yield under forage restriction (42–44). Such coordinated changes underscore a functional microbiome–metabolome interface that bridges host nutrient metabolism and microbial activity under hypoxia. Integrating these correlations with the observed serum biochemical and metabolomic profiles provides mechanistic support for the proposed serum–metabolome–microbiota axis in yak high-altitude adaptation.

Together, our findings delineate a tightly coupled host–microbiota system that enables yaks to maintain physiological equilibrium across altitudinal gradients. The reduction in protein metabolism, suppression of the GH/IGF-1 axis, enhancement of antioxidant defenses, and pro-inflammatory immune adjustment collectively form a host-level energy conservation strategy. Simultaneously, metabolic remodeling through retinol, ALA, and TCA pathways optimizes oxygen utilization and antioxidant capacity. The microbiota mirrors these host adaptations: declines in Bacteroidetes and enrichment

of *Christensenellaceae* and *Monoglobus* reflect a community transition toward efficient fermentation under nutrient scarcity. This integrated framework extends previous observations that grazing and diet influence yak metabolic and microbial profiles by identifying altitude as an additional regulatory force shaping systemic adaptation (8).

The adaptive responses observed in yaks embody a finely tuned trade-off between survival and productivity under chronic hypoxia. Reduced anabolic metabolism and microbial fermentation efficiency conserve energy but limit growth potential at extreme altitudes. These findings highlight potential avenues for nutritional and microbial interventions. Supplementation with protein-rich and antioxidant feeds or probiotic strategies targeting *Christensenellaceae* and *Monoglobus* could enhance nutrient utilization and oxidative resilience in plateau herds (17, 39). Furthermore, selective breeding for individuals exhibiting superior antioxidant capacity, immune balance, and fermentation efficiency could promote herd health and sustainable adaptation. Future research should move beyond correlation toward causal validation of host–microbiota interactions under hypoxia. Controlled translocation or hypobaric simulation experiments, combined with host genomics and functional microbiome assays (e.g., bacterial isolation, metatranscriptomics, or fecal microbiota transplantation), could uncover molecular mechanisms and adaptive biomarkers (31, 40). Investigating how dietary modulation of retinol, ALA, and other key metabolites affects adaptation will provide practical guidance for precision feeding strategies. Collectively, these findings emphasize that yak high-altitude resilience arises from the synergistic regulation of host metabolism and gut microbiota, offering a scientific foundation for improved health management, selective breeding, and sustainable yak production on the Qinghai-Tibet Plateau.

## Conclusion

This study provides an integrated view of how yaks achieve physiological and metabolic adaptation to high-altitude environments. With increasing altitude, yaks exhibited reduced protein metabolism and suppressed growth-related endocrine activity, accompanied by enhanced antioxidant defenses and immune activation. Metabolomic analysis revealed distinct shifts in energy and lipid metabolism, particularly through retinol and α-linolenic acid pathways that support oxidative balance and cellular protection under hypoxia. The gut microbiota also underwent compositional and functional reorganization, favoring taxa associated with efficient fermentation and host metabolic stability. Together, these results indicate that yak adaptation to high altitude is driven by a coordinated host–microbiota response that reallocates energy from growth toward maintenance, antioxidation, and immune resilience. Understanding this serum–metabolome–microbiota network offers valuable insights for developing microbiome- and nutrition-based strategies to enhance yak health, performance, and sustainability on the Qinghai-Tibet Plateau.

## ACKNOWLEDGMENTS

This research was funded by the National Key Research and Development Program of China (2022YFD1302101), the China Agriculture Research System Beef Cattle and Yak Industry Technology System (CARS-37), and the Regional Collaborative Innovation Program in Gesangtang, Linzhou County (QYXTZX-LS2020-01).

Y.X.: Writing—original draft, Data curation. Y.C.: Project administration. Z.C.: Investigation, Methodology. R.C.: Supervision. G.L.: Visualization. D.W.: Resources. D.C.: Formal analysis. J.C.: Supervision. J.K.: Validation. Y.Z.: Writing—review and editing, Funding acquisition. W.B.: Conceptualization, Funding acquisition.

## AUTHOR AFFILIATIONS

[1]Institute of Animal Husbandry and Veterinary Medicine, Xizang Academy of Agricultural and Animal Husbandry Sciences, Lhasa, China

[2]State Key Laboratory of Animal Nutrition and Feeding, Institute of Animal Science, Chinese Academy of Agricultural Sciences, Beijing, China

[3]Precision Livestock and Nutrition Unit, TERRA Teaching and Research Centre, Gembloux Agro-Bio Tech, University of Liège, Gembloux, Belgium

[4]Nie Rong County Agriculture and Animal Husbandry Comprehensive Service Center, Nagqu City, China

[5]Institute of Scientific and Technical Information of Xizang Autonomous Region, Lhasa, China

## AUTHOR ORCIDs

Yining Xie http://orcid.org/0009-0005-8983-7169
Yanbin Zhu http://orcid.org/0000-0002-7273-2409

## AUTHOR CONTRIBUTIONS

Yining Xie, Data curation, Writing – original draft | Yangji Cidan, Project administration | Zhuoma Cisang, Investigation, Methodology | Renzeng Ciwang, Supervision | Guifang Liu, Validation | Dan Wu, Resources | Duoji Cideng, Formal analysis | Jiacuo Chilie, Supervision | Jilam Kang, Validation | Yanbin Zhu, Funding acquisition, Writing – review and editing | Wangdui Basang, Conceptualization, Funding acquisition

## DATA AVAILABILITY

The raw sequencing data are available in the NCBI SRA database under the BioProject accession number PRJNA1254471. For more details, please refer to the project page: ID 1254471—BioProject—NCBI.

## ETHICS APPROVAL

All methods in this study were conducted in strict accordance with the relevant guidelines and regulations. The study followed the ARRIVE Guidelines. Additionally, all procedures, including animal experiments and sample collection, were approved by the Laboratory Animal Welfare and Ethics Committee of the Institute of Animal Husbandry, Tibet Academy of Agricultural and Animal Husbandry Sciences (approval no.: 20240501).

## ADDITIONAL FILES

The following material is available online.

### Supplemental Material

**Fig. S1 (Spectrum02549-25-S0001.tif).** Differential relative abundances of microbial taxa at the phylum and genus levels.
**Supplemental material (Spectrum02549-25-S0002.docx).** Supplemental figure legend.

### Open Peer Review

**PEER REVIEW HISTORY (review-history.pdf).** An accounting of the reviewer comments and feedback.

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
