## [Reviewer comments · Microbiology Spectrum]

Microbiology Spectrum

Effect of altitudes on serum parameters, metabolome, and gut microbiota in yaks on the Qinghai-Tibet Plateau

Yining Xie, Yangji Cidan, Zhuoma Cisang, Ciwang Renzeng, Guifang Liu, Dan Wu, Duoji Cideng, Jiacao Chile, Jilamu Kang, Yanbin Zhu, and Wangdui Basang

Corresponding Author(s): Yanbin Zhu, Gansu Agricultural University

Review Timeline:

Submission Date:	August 17, 2025
Editorial Decision:	September 22, 2025
Revision Received:	October 27, 2025
Accepted:	November 12, 2025

Editor: Hongan Long

Reviewer(s): The reviewers have opted to remain anonymous.

Transaction Report:

DOI: <https://doi.org/10.1128/spectrum.02549-25>

Re: Spectrum02549-25 (**Effect of altitudes on serum parameters, metabolome, and gut microbiota in yaks on the Qinghai-Tibet Plateau**)

Dear Mr. Cideng Duoji:

Thank you for the privilege of reviewing your work. Below you will find my comments, instructions from the Spectrum editorial office, and the reviewer comments.

Revision Guidelines

Sincerely,
Hongan Long
Editor
Microbiology Spectrum

Reviewer #1 (Comments for the Author):

Here are my review comments on the manuscript titled "Effect of altitudes on serum parameters, metabolome, and gut microbiota in yaks on the Qinghai-Tibet Plateau":

I. Research Background and Significance

The manuscript focuses on the adaptive mechanisms of yaks to different altitudinal environments on the Qinghai-Tibet Plateau, investigating multiple aspects including serum biochemistry, antioxidant capacity, immune response, hormones, metabolome, and gut microbiota. The study holds significant scientific and practical importance. It enhances our understanding of how animals adapt to extreme environments and provides valuable theoretical support for yak husbandry in plateau regions, which is crucial for local herders' livelihoods and the sustainable development of the livestock industry.

II. Research Methods

The experimental design is rational. The study selected 30 yaks from three different altitudes for sample collection, covering a comprehensive range of tests from serum analysis to metabolomics and gut microbiota. The sample processing and analysis methods are scientifically sound and well-established. The use of various diagnostic kits, LC-MS/MS technology, and 16S rRNA gene sequencing ensures the reliability and accuracy of the data obtained.

III. Research Results

1. Serum Biochemical Indicators^{**}: With increasing altitude, indicators such as total protein and globulin decreased, while triglycerides showed significant differences between groups. This reflects changes in the nutritional metabolism and liver function of yaks in high-altitude environments.
2. Antioxidant Indicators^{**}: The increase in T-SOD and T-AOC, along with the decrease in GSH-Px, indicates that yaks face higher oxidative stress at higher altitudes and have adapted by enhancing their antioxidant defense mechanisms.
3. Immune Indicators^{**}: The rise in immunoglobulins and pro-inflammatory cytokines, coupled with a decrease in anti-inflammatory cytokines, shows that yaks have activated adaptive immune regulation to cope with potential infections and tissue damage in high-altitude areas.
4. Hormonal Indicators^{**}: The decline in GH, IGF-1, and GHRH, along with an increase in GHIH, aligns with the strategy of prioritizing survival over growth under hypoxic conditions, which is a common adaptive response in high-altitude environments.
5. Metabolome and Gut Microbiota^{**}: Significant changes in plasma metabolites and fecal microbiota structure were observed with increasing altitude. These changes, such as alterations in specific metabolic pathways and shifts in microbial abundance, highlight how yaks reprogram their metabolism and adjust their gut microbiota to adapt to high-altitude conditions, which in turn affects their energy metabolism and overall health.

IV. Conclusions

The manuscript concludes that yaks employ a multi-systemic approach to adapt to high-altitude environments, involving enhanced antioxidant and immune responses, hormone-mediated growth suppression, metabolic reprogramming, and gut microbiota restructuring. These adaptations enable yaks to survive and conserve energy under hypoxic stress. The findings are significant for advancing our knowledge of animal adaptation to extreme environments and have practical implications for yak breeding and health management in plateau areas.

V. Suggestions

1. Consistency in Formatting^{**}: The authors should ensure consistency in the formatting of statistical notations, such as using italic for "P" values (line 30) and other statistical symbols throughout the manuscript.
2. Abbreviations^{**}: All abbreviations, including T-SOD, T-AOC, GSH-Px, IL-2, IL-6, etc., should be defined in full at their first appearance, especially in the abstract, to ensure clarity for readers who may not be familiar with these terms.
3. Uniform Indentation: The manuscript should adhere to the journal's formatting guidelines for indentation (line 105). This includes uniform indentation in the main text and other sections to maintain a professional and consistent appearance.
4. Consistent Capitalization: The authors should ensure consistent capitalization in subheadings (line 166) and throughout the manuscript to avoid confusion and maintain a polished presentation.
5. Future Research Directions: The discussion section could benefit from a forward-looking perspective on the long-term adaptations of yaks to high-altitude environments. For example, it would be valuable to speculate on whether these adaptive changes might have long-term impacts on the reproductive performance, growth, and meat quality of yaks. Additionally, suggestions for scientific breeding practices that could further optimize the high-altitude adaptability of yaks would enhance the practical applicability of the study.

Reviewer #2 (Comments for the Author):

This manuscript presents a comprehensive multi-omics investigation into how yaks adapt to different altitudes on the Qinghai-Tibet Plateau, examining serum biochemistry, antioxidant status, immune function, hormone profiles, plasma metabolome, and gut microbiota. The integration of physiological, metabolomic, and microbial data provides a holistic view of adaptation strategies in an important livestock species. The topic is timely, relevant to both animal physiology and microbiome research, and fits well within the scope of Microbiology Spectrum, which welcomes studies combining microbiology with broader host physiology and omics approaches.

The study is generally well designed, the data are robust, and the results are of potential interest to readers in microbiology, animal science, and high-altitude biology. However, there are several issues that need to be addressed before the manuscript can be considered for publication.

Major Comments

1. While the study integrates multiple layers of data, the novelty compared to existing yak adaptation studies (including microbiome-focused works) should be more clearly articulated. Some of the findings (e.g., altered antioxidant enzyme activity, reduced VFAs, or immune changes at altitude) are consistent with previously reported trends in other high-altitude animals. The authors should emphasize what is truly new here (e.g., the integrative view across serum, metabolome, and microbiota; specific pathways such as retinol and α -linolenic acid metabolism). A clearer positioning against recent studies (including their own related Spectrum paper, ref. 18) is needed.
2. The study includes only 10 yaks per altitude group (n=30 total). While this is understandable given logistical challenges, the statistical power may be limited. The authors should explicitly discuss this limitation and justify whether the conclusions (especially for multi-omics correlations) remain robust. Clarification on whether animals were matched for age, diet, and management is also necessary, since these confounders can strongly influence microbiota and metabolome profiles.
3. Although the manuscript presents data from serum, metabolome, and microbiota separately, the integration across these layers remains relatively descriptive. To strengthen the manuscript, the authors should explore correlations or network analyses linking metabolite pathways with specific microbial taxa and serum parameters. This would provide stronger evidence for functional host-microbiota interactions underlying altitude adaptation.
4. The microbiota analysis is presented mainly at genus level with alpha/beta diversity and LEfSe. However, given the importance of microbial function in high-altitude adaptation, predictive functional analyses (e.g., PICRUSt2, KEGG module associations) should be reported more thoroughly. Currently, only BugBase predictions are briefly described, which feels insufficient. Strengthening this part would make the manuscript more attractive to a microbiology-focused readership.
5. The discussion is well written but remains somewhat speculative in parts. For example, linking reduced VFAs directly to nutrient availability or growth inhibition requires stronger evidence. The authors should be careful to distinguish between supported findings and hypotheses, and potentially suggest future directions for experimental validation (e.g., controlled feeding studies or metabolite supplementation trials).

Minor Comments

1. The abstract is informative but overly dense. Consider shortening sentences and highlighting the most novel findings rather than listing too many details.
2. Figures are generally clear, but some (e.g., PCA, volcano plots, LEfSe trees) could benefit from larger font sizes and simplified legends to improve readability.
3. The Introduction is comprehensive but somewhat lengthy. Streamlining the background would help maintain focus on the study's central hypothesis.
4. References are generally appropriate, but the manuscript could benefit from citing a few more recent microbiome-altitude studies in livestock and humans to contextualize the findings.
5. Minor language issues exist throughout (e.g., "the enzymatic activity of GSH-Px were decreased" → "was decreased"). A careful proofreading or professional English editing is recommended.
6. The "Importance" section is suitable for Spectrum, but could be made more concise and targeted to the microbiology readership by emphasizing microbial contributions to adaptation rather than broader physiology.

This manuscript presents a comprehensive multi-omics investigation into how yaks adapt to different altitudes on the Qinghai-Tibet Plateau, examining serum biochemistry, antioxidant status, immune function, hormone profiles, plasma metabolome, and gut microbiota. The integration of physiological, metabolomic, and microbial data provides a holistic view of adaptation strategies in an important livestock species. The topic is timely, relevant to both animal physiology and microbiome research, and fits well within the scope of *Microbiology Spectrum*, which welcomes studies combining microbiology with broader host physiology and omics approaches. The study is generally well designed, the data are robust, and the results are of potential interest to readers in microbiology, animal science, and high-altitude biology. However, there are several issues that need to be addressed before the manuscript can be considered for publication.

Major Comments

1. While the study integrates multiple layers of data, the novelty compared to existing yak adaptation studies (including microbiome-focused works) should be more clearly articulated. Some of the findings (e.g., altered antioxidant enzyme activity, reduced VFAs, or immune changes at altitude) are consistent with previously reported trends in other high-altitude animals. The authors should emphasize what is truly *new* here (e.g., the integrative view across serum, metabolome, and microbiota; specific pathways such as retinol and α -linolenic acid metabolism). A clearer positioning against recent studies (including their own related *Spectrum* paper, ref. 18) is needed.
2. The study includes only 10 yaks per altitude group (n=30 total). While this is understandable given logistical challenges, the statistical power may be limited. The authors should explicitly discuss this limitation and justify whether the conclusions (especially for multi-omics correlations) remain robust. Clarification on whether animals were matched for age, diet, and management is also necessary, since these confounders can strongly influence microbiota and metabolome profiles.
3. Although the manuscript presents data from serum, metabolome, and microbiota separately, the integration across these layers remains relatively descriptive. To strengthen the manuscript, the authors should explore correlations or network analyses linking metabolite pathways with specific microbial taxa and serum parameters. This would provide stronger evidence for functional host–microbiota interactions underlying altitude adaptation.
4. The microbiota analysis is presented mainly at genus level with alpha/beta diversity and LEfSe. However, given the importance of microbial function in high-altitude adaptation, predictive functional analyses (e.g., PICRUSt2, KEGG module associations) should be reported more thoroughly. Currently, only BugBase predictions are briefly described, which feels insufficient. Strengthening this part would make the manuscript more attractive to a microbiology-focused readership.
5. The discussion is well written but remains somewhat speculative in parts. For example, linking reduced VFAs directly to nutrient availability or growth inhibition requires stronger evidence. The authors should be careful to distinguish between supported findings and hypotheses, and potentially suggest future directions for experimental validation (e.g., controlled feeding studies or metabolite supplementation trials).

Minor Comments

1. The abstract is informative but overly dense. Consider shortening sentences and highlighting the most novel findings rather than listing too many details.
2. Figures are generally clear, but some (e.g., PCA, volcano plots, LEfSe trees) could benefit from larger font sizes and simplified legends to improve readability.
3. The Introduction is comprehensive but somewhat lengthy. Streamlining the background would help maintain focus on the study's central hypothesis.
4. References are generally appropriate, but the manuscript could benefit from citing a few more recent microbiome–altitude studies in livestock and humans to contextualize the findings.
5. Minor language issues exist throughout (e.g., “the enzymatic activity of GSH-Px were decreased” → “was decreased”). A careful proofreading or professional English editing is recommended.
6. The "Importance" section is suitable for *Spectrum*, but could be made more concise and targeted to the microbiology readership by emphasizing microbial contributions to adaptation rather than broader physiology.

Dear Editor and Reviewers,

We sincerely thank you for your valuable time and insightful comments on our manuscript. We are very grateful for the constructive feedback provided, which has greatly helped us improve the clarity, rigor, and overall quality of our work. We have carefully revised the manuscript in accordance with all suggestions. Below, we provide a detailed, point-by-point response to each comment. Revised portions in the manuscript are indicated with corresponding line numbers.

We hope that the revised version meets your expectations and sincerely appreciate your consideration.

Response to Reviewer 1

1. Consistency in Formatting

Comment: The authors should ensure consistency in the formatting of statistical notations, such as using italic for "P" values (line 30) and other statistical symbols throughout the manuscript.

Response: We have carefully standardized all statistical notations throughout the manuscript. For example, $P < 0.05$ (line 32) has been italicized, and the same format has been applied consistently across the entire text.

2. Abbreviations

Comment: All abbreviations, including T-SOD, T-AOC, GSH-Px, IL-2, IL-6, etc., should be defined in full at their first appearance, especially in the abstract.

Response: We have revised the manuscript to ensure that all abbreviations are clearly defined at their first appearance, particularly in the Abstract and Introduction, to enhance readability and accessibility for all readers.

3. Uniform Indentation

Comment: Ensure consistent indentation throughout the manuscript.

Response: We have reformatted the manuscript according to the journal's style guide to maintain uniform indentation and alignment (e.g., see line 175).

4. Consistent Capitalization

Comment: Ensure consistent capitalization in subheadings (line 166).

Response: All subheadings have been standardized to the same capitalization style (first letter capitalized, remaining lowercase), in line with the journal's requirements (see line 181).

5. Future Research Directions

Comment: Add a forward-looking perspective on long-term yak adaptations (e.g., reproduction, growth, meat quality) and potential breeding strategies.

Response: We have expanded the Discussion (lines 626–645) to include perspectives on long-term adaptive effects on reproductive performance, growth, and meat quality. We also added suggestions for scientific breeding strategies to further optimize yak adaptability, enhancing the practical relevance of the study.

Response to Reviewer 2

Major Comments

1. Novelty and Positioning

Comment: The novelty of the study compared to existing yak adaptation studies should be more clearly articulated.

Response: We have revised the Discussion to highlight the novelty of this study. Specifically, we emphasize our integrated multi-omics approach linking serum biochemistry, metabolome, and microbiota. We also discuss the discovery of altitude-responsive pathways such as retinol and TCA cycle metabolism, which extend beyond previous yak studies, including our own Spectrum paper (ref. 18).

2. Sample Size and Statistical Robustness

Comment: The limited sample size ($n = 10$ per group) may affect statistical power. Please justify and clarify controls for confounders.

Response: We acknowledge the limitation of the sample size ($n = 10$ per group, total $n = 30$), which may reduce the ability to detect small effects. However, this sample size is consistent with prior high-altitude ruminant studies, where logistical and ethical constraints limit larger sampling. To ensure robustness, we employed rigorous data quality control — excluding variables with $>20\%$ missing values, applying normalization, conducting seven-fold cross-validation, and selecting features based on $VIP > 1$ and $P < 0.05$. All yaks were 2.5-year-old heifers, sampled in the same season (September 2024), grazing naturally under identical management conditions. These measures minimized confounding factors such as age, diet, and management. The consistent multi-omics trends observed across datasets support the reliability and biological significance of our findings.

3. Integration Across Omics Layers

Comment: Integration across serum, metabolome, and microbiota

remains largely descriptive.

Response: We have added correlation and network analyses linking metabolites, microbial taxa, and serum biochemical indices (lines 435–457). Moreover, the Discussion has been expanded to highlight the mechanistic connections among these omics layers in the context of altitude adaptation.

4. Microbial Functional Analysis

Comment: Functional predictions (e.g., PICRUSt2, KEGG modules) should be discussed in more detail.

Response: We have added detailed results from PICRUSt2-based functional prediction and KEGG module enrichment (lines 397–404), and further elaborated on these microbial functional shifts in the Discussion.

5. Speculative Discussion and Future Validation

Comment: The discussion should better distinguish between evidence-based conclusions and hypotheses, with suggestions for experimental validation.

Response: We have revised the Discussion to clearly separate supported results from hypotheses. In addition, we now propose future research directions involving controlled feeding and metabolite supplementation experiments to validate the hypothesized adaptive mechanisms.

Minor Comments

1. Abstract Clarity

Response: We have condensed and refined the Abstract (lines 22–45), highlighting key novel findings and reducing descriptive detail.

2. Figure Readability

Response: All figures were reformatted with larger fonts and simplified legends for improved readability (lines 1124–1151).

3. Introduction Length

Response: The Introduction has been streamlined to better focus on the study's core hypothesis and objectives.

4. References

Response: We have added several recent references (e.g., Ma et al., 2025; Xie et al., 2025) to strengthen the context of high-altitude microbiome research.

5. Language Editing

Response: We have carefully proofread the entire manuscript to correct minor grammatical inconsistencies (e.g., “were decreased” → “was decreased”) and ensure language accuracy.

6. Importance Section

Response: We have rewritten the “Importance” section to focus on microbial contributions to yak high-altitude adaptation, aligning it with *Microbiology Spectrum*'s readership (line 71-83).

In summary, we sincerely appreciate the reviewers' constructive feedback and have addressed all comments with corresponding revisions. We believe these improvements have substantially enhanced the clarity, depth, and impact of our study.

Thank you again for your valuable input and kind consideration.

Yining Xie

Re: Spectrum02549-25R1 (**Effect of altitudes on serum parameters, metabolome, and gut microbiota in yaks on the Qinghai-Tibet Plateau**)

Dear Dr. yanbin Zhu:

Your manuscript has been accepted, and I am forwarding it to the ASM production staff for publication. Your paper will first be checked to make sure all elements meet the technical requirements. ASM staff will contact you if anything needs to be revised before copyediting and production can begin. Otherwise, you will be notified when your proofs are ready to be viewed.

Sincerely,
Hongan Long
Editor
Microbiology Spectrum

Reviewer #1 (Comments for the Author):

None.

Reviewer #2 (Comments for the Author):

All my previous comments have been adequately addressed. I recommend the manuscript for acceptance.

All my previous comments have been adequately addressed. I recommend the manuscript for acceptance.